# Knowledge and Attitudes of Guam Residents towards Cancer Clinical Trial Participation

**DOI:** 10.3390/ijerph192315917

**Published:** 2022-11-29

**Authors:** Munirih R. Taafaki, Amy C. Brown, Kevin D. Cassel, John J. Chen, Eunjung Lim, Yvette C. Paulino

**Affiliations:** 1Department of Quantitative Health Sciences, John A. Burns School of Medicine, University of Hawaiʻi at Mānoa, Honolulu, HI 96813, USA; 2Population Sciences in the Pacific Program, University of Hawaiʻi Cancer Center, Honolulu, HI 96813, USA; 3School of Health, University of Guam, Mangilao, GU 96923, USA

**Keywords:** cancer, clinical trial, Guam, knowledge, attitudes

## Abstract

(1) Background: Currently there are no cancer clinical trials in Guam, where CHamoru people suffer the highest rates of cancer mortality, and interest to do so is growing. This study investigated the knowledge and attitudes of Guam residents towards cancer clinical trial participation prior to implementation. (2) Methods: A telephone survey was developed, tested, and conducted among Guam resident adults, 18 years of age and older. Survey questions were summarized by descriptive statistics. Logistic regression models were used to investigate the associations between Guam residents’ demographics and their clinical trial knowledge and attitudes. Adjusted odds ratios (aOR) and associated 95% confidence intervals (CI) were calculated. (3) Results: One hundred fifty-two people participated in the survey, most of whom were CHamoru (47.0%). Fifty-three percent had heard the term ‘clinical trial’; 73.7% would take part in a trial if they had cancer; and 59.9% believed they would receive good quality treatment from a trial offered in Guam. CHamoru were more likely than Whites to associate out-of-pocket expenses with clinical trial participation (aOR = 5.34, 95% CI = 1.68–17.00). Physician ethnicity was important to 30% of non-Whites and significantly associated with those who spoke a language other than English (aOR = 3.40, 95% CI = 1.29–8.95). Most people (65.0%) did not believe clinical trials participants were ‘guinea pigs’. (4) Conclusion: Though knowledge about cancer clinical trials is limited, attitudes were primarily positive towards participating in cancer clinical trials offered in Guam. Future delivery of cancer clinical trials will benefit from identifying potential barriers to recruitment and adopting an approach suited to Guam’s population.

## 1. Introduction

Cancer clinical trials are among the most effective methods of informing medical practice. However, accrual onto cancer clinical trials across the United States (U.S.) is low, and the rate of minority enrollment is disproportionate to both the representation of minorities in the U.S. and the national percentage of minorities burdened with cancer [1]. The National Institutes of Health’s (NIH) Revitalization Act mandates inclusion of women and minorities in federally funded clinical research and underscores the importance of making clinical trials available for underrepresented populations to improve health [2]. Pacific Islanders are one of the fastest-growing ethnic minority in the U.S. [3,4] and by 2030, an over 100% increase in cancer incidence for many cancer sites is projected for this population [5]. Without the participation of minorities in cancer clinical trials, including Pacific Islanders, the development of effective interventions in diverse populations is limited [6,7]. Therefore, research is needed to understand knowledge and attitudes that impact Pacific Islanders’ decision to participate in cancer clinical trials and increase their enrollment.

Interest in introducing cancer clinical trials in Guam has grown because amidst expanding screenings and treatments, cancer continues to be its second highest cause of death [8]. Data collected between 1998–2012 suggests cancer incidence and mortality rates in Guam are increasing and vary across ethnic groups [9]. CHamoru, Guam’s indigenous and largest ethnic population, suffer the highest incidence and mortality rates [10,11]. Guam’s ethnically diverse population provides a unique setting for researchers and medical providers to learn about health needs of Pacific Islanders, who are underserved and underrepresented [4].

Knowledge and attitude towards clinical trials plays an important role in the decision to enroll [7]. Known barriers of cancer clinical trial participation among ethnic minorities include lack of knowledge and awareness about clinical trials [1], attitudes towards cost including out-of-pocket expenses [12,13,14] and health insurance coverage [13,14,15,16], medical and physician distrust and fear of experimentation [1,2,7,12,13,15,16,17,18,19,20], cultural factors such as attitude towards physician ethnicity [2], family, religious and community support, language, and access to cancer clinical trials [3,21,22,23].

To date, cancer clinical trials have not been offered in Guam and information on the knowledge and attitudes of its population towards such trials is limited. Using a population-based telephone survey, this study aimed to acquire a preliminary understanding of Guam residents’ knowledge and attitudes towards cancer clinical trials based on socio-demographics including ethnicity, income, education, gender, language, religion, and health insurance to gauge potential barriers to recruitment onto cancer clinical trials among Guam’s diverse community.

## 2. Materials and Methods

### 2.1. Survey Design and Development

A 43-item telephone survey, comprised of 12 demographic and 31 clinical trial knowledge and attitude questions, was conducted in Guam. Conducting the survey via telephone was the decidedly preferred method in consultation with third-party marketing company, Anthology Group, which has 15 years of experience conducting survey studies in Guam, is knowledgeable of the region’s high rate of telephone ownership, and helped to facilitate the survey. The survey was offered in English, one of its official languages (the other is CHamoru). CHamoru word definitions were provided as needed [24]. Questions were created to address concerns specific to introducing cancer clinical trials in Guam including preferences to leave or stay on island to receive treatment, beliefs about culture, benefits, and quality of treatment by participating in a clinical trial offered in Guam. Other items were adapted from established surveys to evaluate factors affecting knowledge and attitude about cancer clinical trials related to awareness [14,16,21], health insurance coverage and financial cost [14,16], and the fear of experimentation [13,16], and known concerns for indigenous populations including cultural beliefs and physician ethnicity [25]. Of the 31 knowledge and attitude survey questions, several were specific to knowledge and awareness of clinical trials (Knowledge 1–3, Attitudes 5–7) (Appendix A). Others aimed to gauge understanding of clinical trial-related costs and insurance coverage (Knowledge 4, Attitudes 16–17); benefits and risks of participation (Attitudes 10–12, 15); and mistrust of experimentation and Western medicine (Attitudes 14, 23). Cultural attitudes towards cancer and clinical trials were sought through questions about physician ethnicity (Attitudes 20–21), Western medical advice perceived as contrary to cultural beliefs (Attitude 22), and family, social (Attitude 24), religious and community support (Attitude 25).

The survey questions used a 3-point Likert scale with the option to refuse an answer. Questions about knowledge and attitudes towards clinical trials and cultural factors preceded multiple demographic questions. Demographic variables collected were: age, ethnicity, marital status, education, personal income, born country, employment status, English fluency, whether speaking other language or not, healthcare coverage, insurance type, and religion.

The survey was pilot-tested in Guam using two focus groups comprised of 34 residents. Responses showed a good reliability (Cronbach’s alpha = 0.73). The survey was then further modified for functionality. Approval of the study was obtained from the Institutional Review Boards of the University of Hawaiʻi and the University of Guam.

### 2.2. Recruitment and Data Collection

Inclusion criteria were adults residing in Guam at least 18 years old and able to speak and understand English.

Recruitment and telephone interviews were conducted by Anthology Group during Guam peak times in October 2018. Proprietary random digit dialing software (Anthology Marketing, Honolulu, HI, USA) was utilized due to its ability to select telephone numbers at random, broaden sample selection, locate numbers which may be missing from a phone book, and provide complete coverage of a geographic region with high telephone ownership rates. Up to three attempts were made to reach each working telephone number. The study sample comprised of respondents who completed the survey via landline or mobile telephones.

### 2.3. Data Analysis

Data analysis was restricted to completed survey responses only. Certain demographic categories, including ethnicity, education, income, employment, health insurance, and marital status, were combined based on the distribution of responses for each variable to facilitate the data analysis. CHamoru, Filipino, White and ‘Other’ were the key ethnicity categories. Education was coded into three levels: high school graduate equivalence or less, some college or technical school, and equal to college graduate or higher. Personal income was grouped into less than $50,000, $50,000 or more, and ‘Refused’. ‘Refused’ to respond was treated as missing for all variables except personal income, where it was kept as a separate category due to the number of individuals who refused to disclose their range of income. Employment status was categorized into full-time or part-time, retired, and unemployed/student/homemaker/unable to work. Type of insurance was categorized as ‘Private’, ‘Public’ (Medicare, Medicaid, military), and ‘No Insurance.’ Marital status was categorized as married or living as married, single, and divorced/widowed.

Participants’ demographics were summarized by descriptive statistics: means, standard deviations, and ranges for continuous variables; frequencies and percentages for categorical variables. Answers to each Likert scale survey question was dichotomized and treated as a binary outcome (Positive vs. Negative). ‘Yes’ and ‘strongly or somewhat agree’ were treated as ‘Positive’ and ‘No or Not sure’ and ‘strongly or somewhat disagree’ were treated as ‘Negative’ depending on the answer choices. For robustness of the study findings, multivariable data analyses were restricted for those survey questions where both of the combined categories of ‘Positive’ and ‘Negative’ comprised at least 30% of the total sample.

Univariate logistic regression analyses were conducted to investigate bivariate associations between knowledge and attitude survey questions and demographic variables. Multivariable logistic regression models were then developed considering all variables of *p*-value < 0.20 in the univariate logistic analysis using a stepwise selection method to identify the set of significant associated demographic variables for each knowledge and attitude survey question. To facilitate the comparison across all knowledge and attitude questions, the final variable set included all variables that were significantly associated with at least one survey question. Adjusted odds ratios (aORs) and their 95% confidence intervals (CIs) were determined. The final logistic models were assessed using Hosmer and Lemeshow goodness of fit tests and c-statistics. All analyses were conducted in SAS version 9.4 (Cary, NC, USA) and two-tailed *p*-value < 0.05 was considered statistically significant.

## 3. Results

Fieldwork resulted in the completion of 152 interviews with a total response rate of 11.4%. Table 1 summarizes demographic characteristics. More females (53.3%) were represented in the study. CHamoru, Guam’s largest ethnic group, accounted for the majority of study participants (47.0%); followed by Filipinos (26.5%); ‘Other’ respondents—comprised of other Pacific Islanders and Asians (15.2%); and Whites (11.3%). Forty-nine percent were college graduates and 48.0% reported earning an annual income of less than $50,000, with 11.2% of the respondents refusing to answer. Most survey respondents said they spoke English well (91.3%) and reported that they also spoke a language other than English (70.7%). Most had private insurance (59.3%), followed by 27.6% with public insurance, and 13.1% with no insurance.

Response rates for the Knowledge and Attitude questions are listed on Table 2. More than half of the respondents had heard of the term ‘clinical trial’ (58.6%) (Knowledge 1). Univariable logistic regression analyses revealed significant associations of this question with ethnicity, education, income, country of birth, health insurance and language variables (Appendix A). For education, the odds of having heard of the term increased 5.04 times for college graduates, compared to high school graduates or less. Nearly 70% knew participating in a clinical trial may mean not receiving the treatment being tested (Knowledge 3). Only 32.9% agreed that in a clinical trial the sponsor pays for the study drug while all other costs are billed to insurance (Knowledge 4). For this question, education, employment, language (speaking English and/or another language), and insurance coverage and type were significantly associated in univariable logistic regression analyses (Appendix A).

Nearly sixty percent thought they would receive good quality treatment from a clinical trial offered in Guam (Attitude 13). Fewer Guam respondents (34.9%) thought cancer clinical trials participants are treated like guinea pigs (Attitude 14). More participants (56.0%) thought they would pay more out-of-pocket expenses if they took part in a clinical trial in Guam (Attitude 17). The majority of respondents reported physician ethnicity (Attitude 20) would not be important in their decision to take part in a cancer clinical trial (69.1%). However, 30% or more CHamoru, Filipino and ‘Other’ said it was important. Univariable logistic regression analyses found significant associations with education, income, place of birth, and the language and insurance variables.

Twelve knowledge and attitude survey questions met the multivariable data analytical criterion. Results for the multivariable logistic regressions suggest that questions on clinical trial knowledge were most closely associated with education, income, employment, language (English fluency and speaking another language), or health insurance (Table 3). More likely to have heard the term ‘clinical trial’ were college graduates (aOR = 4.43, 95% CI = 1.97–9.95) than those with a high school education or less; and individuals who spoke English well compared to those who did not (aOR = 6.02, 95% CI = 1.21–29.88). College graduates (aOR = 0.23, 95% CI = 0.07–0.74) and those who spoke English well (aOR = 0.04, 95% CI = 0.01–0.25) were less likely to be disagree regarding new drug and other cost coverage during clinical trial participation (Knowledge 4). Results by insurance coverage and type show that in comparison to private insurance holders, those with public insurance (aOR = 7.51, 95% CI = 2.65–21.31) and no insurance (aOR = 6.90, 95% CI = 1.85–25.71) were significantly more likely to agree, as were those whose annual income was $50,000 or higher (aOR = 6.53, 95% CI = 1.81–23.57) or refused to disclose income information (aOR = 9.00, 95% CI = 1.84–44.11) compared with those earning $50,000 or less.

There were less obvious patterns across the clinical trial attitude survey questions analyzed (Table 3). Ethnicity was related with Attitude 17 and Attitude 25 where CHamoru were more likely than Whites (aOR = 5.34, 95% CI = 1.68–17.00) to think they would need to pay out-of-pocket to participate in a clinical trial (Attitude 17); and compared to Whites, more CHamoru (aOR = 27.70, 95% CI = 3.47–221.00) and Filipinos (aOR = 42.20, 95% CI = 4.98–357.00) considered religious community support very or somewhat important in their decision to participate (Attitude 25).

Education remained statistically significant in a multivariable logistic model with attitudes towards physician ethnicity (Attitude 20) and medical advice contrary to cultural beliefs (Attitude 22). Most college educated respondents were less likely to think physician ethnicity mattered compared to high school graduates or less (aOR = 0.34, 95% CI = 0.15–0.74); and more likely to listen to medical advice that went against cultural beliefs (aOR = 3.42, 95% CI = 1.17–10.00). Those speaking a language other than English were more likely to think physician ethnicity was important (aOR = 3.40, 95% CI = 1.29–8.95). For country of birth, Philippines-born was most associated with Attitude 14 where those individuals were six times more likely than U.S.-born (aOR = 6.14, 95% CI = 1.84–20.50) to think clinical trial participants are treated like guinea pigs. Income remained statistically significant for Attitude 13 where individuals with a $50,000 annual income or more (aOR = 2.44, 95% CI = 1.18–5.02) were more likely to agree that they would receive good quality treatment from a clinical trial offered in Guam than those making less.

## 4. Discussion

To our knowledge, this is the first study to determine the knowledge and attitudes of Guam’s population towards cancer clinical trial participation. The demographic of the survey sample shared several similarities to Census Bureau [26] and Pew Research Center [27] data for ethnicity, education, employment status, health insurance, income, and religion. The percentage of CHamoru and Whites in the study were proportionally higher than are present in Guam’s population. In contrast, fewer Other respondents took part in the study than make up Guam’s residents, which include not only other Pacific Islanders and Asians, but also Black, Hispanic, other, and multiple ethnic origins. The rate of study participants with a college or graduate degree exceeded the overall rate of individuals in Guam with that level of education. Survey respondents reflected a higher percentage of those privately and publicly insured and a lower percentage of uninsured compared to Guam’s 2010 U.S. Census records, which indicate 49.1% of residents carry private health insurance, 22.4% carry public insurance, and 7.4% carry both types (a category the survey did not ask for). Twenty-one percent of Guam’s residents are uninsured [26]. This rate has not altered significantly since 2004 U.S. Census Bureau data [28].

Knowledge responses indicated that most survey respondents had heard about clinical trials but did not know what they are. After a definition of clinical trials was provided nearly three-fourths said they would participate if they had cancer. Results were comparable to but slightly lower than Lara et al.’s study in which participants were cancer patients (and their relatives and friends) [16]. Guam participants aware of clinical trials were primarily college educated and spoke English well. Though most of Guam’s population is proficient in English, most of its population does not have a college education [26]. This suggests most individuals in Guam may not be knowledgeable about clinical trials, which is supported by other studies that report levels of clinical trial awareness decrease when associated with minority ethnicities, lower education levels, and language barriers [1,2,7,12,13,21,29].

Our study showed most respondents thought they would benefit from and receive good quality treatment from a clinical trial offered in Guam. This is encouraging since transportation and distance to clinical trial sites are identified as barriers to accrual [12]. Travel cost to and from Guam, Hawai‘i, and other parts of the U.S. is extremely expensive. Few airlines fly between these locations, and only one offers direct flights. Housing in the treatment location could also be cost-prohibitive. As Guam’s median family and non-family annual incomes are at, or less than, $50,000, respectively [26], willingness to participate in a cancer clinical trial outside of Guam may be a serious concern for more than half its population.

Health insurance coverage results support findings from previous studies which suggest concern and fear of insurance denial is among the highest rated barriers to cancer clinical trials [12,16]. Most respondents disagreed or were not sure clinical trial costs are covered by clinical trial sponsors and insurance. Many of these individuals were publicly insured or had no insurance. Nearly a quarter of Guam’s residents have no health insurance coverage [26,28]. While difficult to draw conclusions from these results, it is possible many Guam residents may perceive health insurance coverage as a financial barrier.

Further related to cost, more than half the respondents thought they would have to pay out-of-pocket expenses. Several studies have cited out-of-pocket costs for clinical trial treatment and the perceived cost of cancer care as barriers to participation [12,14]. Direct and indirect costs have been cited as the third most common barrier to acceptance of enrollment [7]. Our finding indicates CHamoru respondents were concerned about personal financial costs for clinical trials, which relates to findings reported by Moss (2013) that financial barriers to cancer treatment for CHamoru were significant and delayed or missed cancer screenings could be due to cost burden [28]. As patients may be open to explanations on expenses before trial enrollment [14] discussing this topic openly with patients would help clarify and mitigate concern about cost and health insurance coverage.

A common perception of clinical trials is that they are medical experiments done without patients’ knowledge and not for their benefit or safety, and that the participants are ‘guinea pigs’ [17]. Mistrust of Western medical providers and researchers is a recognized barrier to clinical trial participation among indigenous populations. In a Hawaiʻi study measuring the knowledge, attitudes and practices of primary care physicians (PCP) related to cancer screening and cancer prevention clinical trials, 5.0% of PCPs reported their patients did not want to be “research guinea pigs” and 10.0% acknowledged Native Hawaiians’ medical mistrust [30]. In our study, fewer respondents thought cancer clinical trials participants are treated like guinea pigs, compared to participants in Comis et al.’s (2003) nation-wide study [14]. However, those born in the Philippines were more likely to think this about trial participants. Since a sizeable portion of Guam’s population are Filipinos and/or born in the Philippines, it is important to determine if fear of experimentation could influence their decision to enroll onto cancer clinical trials.

Cultural factors also yielded informative results. Ethnicity demonstrated a thought-provoking distinction as all Whites said physician ethnicity was not important when deciding to participate in a cancer clinical trial but at least one-third of non-White respondents placed importance on this factor. Several studies have cited non-White ethnic groups prefer physicians from their own cultural backgrounds. Reasons include that physicians from other cultures do not show them respect [30,31]; and that medical providers lack culturally appropriate training in communicating with patients [18]. The study from which our survey’s physician ethnicity question was derived stated most of its indigenous participants were willing to take part if the researcher was of the same descent [25]. Our study indicated non-Whites are impacted by language barriers, as described by other studies [22,23]. Significant association with language and education suggests that a large portion of Guam’s population would place importance on physician ethnicity and demonstrates the importance of involving culturally relatable researchers and medical providers, on-call translators, and translated materials. Further, medical providers of a similar demographic have been found to be more interested in serving the needs of their communities [30]. In Guam’s diverse setting, the importance of researchers and providers who are culturally familiar and/or competently trained in the cultures of communities they serve will be important in the recruitment to ethnic minorities.

Religious community support when participating in a cancer clinical trial was most important to Filipinos, CHamoru, and those born in the Philippines and Guam. In Pacific Island cultures, family and religious community support can be important and influential in healthcare decision-making. In CHamoru culture, family relationships and faith are deeply valued and connected to one’s health [32]. A study on Guam breast cancer survivors noted that family, social, and church support were the most important and prevalent sources of support [33]. As a large percentage of Guam residents follow a religion [27], religious community support may be important to most Guam residents.

There were several limitations to this study. Identifying a suitable pre-existing validated survey was challenging. By reviewing and implementing select questions from other surveys, their respective measurement scales could not be applied to the whole survey, which proved challenging for data analyses. Thus, using a self-developed survey with unvalidated measures was a major limitation and the results should be treated with caution. Compared to Guam’s population, more females were represented in the study than males. This may be related to females’ willingness to take time to participate and/or their proactivity in self-care compared to men, and may have skewed results toward more positive responses. Conducting telephone survey required questions with short-answer responses and excluded Guam residents without phones. The small sample size required may not be generalizable to the whole Guam population. Finally, the terms ‘guinea pig’ and ‘ethnicity’ were not defined and their meanings may have been unclear to some.

Despite the above limitations, to our knowledge, this is the first study investigating the knowledge and attitudes of Guam residents towards cancer clinical trial participation. It provides critical insights in informing medical providers and researchers of potential barriers to recruitment onto cancer clinical trials in Guam. The results will also likely contribute to the cultural competence of those implementing the trials among Guam’s diverse community.

## 5. Conclusions

This study supports the notion that difference in knowledge and attitudes exist towards cancer clinical trial participation among Guam’s diverse population and vary by ethnicity, religion, and socio-economic status in Guam. Though knowledge about cancer clinical trials is limited, attitudes were primarily positive towards participating in trials offered in Guam. Cancer clinical trials are not yet currently available in Guam. By understanding these factors and adopting an informed approach suited to Guam’s population, future delivery of cancer clinical trials may benefit.

## Figures and Tables

**Table 1 ijerph-19-15917-t001:** Summary of Study Participant Demographic Characteristics.

Variable	*n* (%)
Age, Mean ± SD (Range)	49.6 ± 14.9 (18–78)
Gender	
Male	71 (46.7%)
Female	81 (53.3%)
Ethnicity	
White	17 (11.3%)
CHamoru	71 (47.0%)
Filipino	40 (26.5%)
Other	23 (15.2%)
Marital Status	
Married or living as married	92 (61.3%)
Single	39 (26.0%)
Divorced/widowed	19 (12.7%)
Education	
≤High school graduate	53 (35.6%)
Some college or technical school	23 (15.4%)
≥College graduate	73 (49.0%)
Personal Income	
Less than $50,000	73 (48.0%)
$50,000 or more	62 (40.8%)
Refused	17 (11.2%)
Born Country	
USA	26 (17.5%)
Guam	75 (50.3%)
Philippines	32 (21.5%)
Other	16 (10.7%)
Employment Status	
Employed full time or part-time	94 (62.7%)
Retired	26 (17.3%)
Unemployed/student/homemaker/unable to work	30 (20.0%)
English Fluency	
Well	137 (91.3%)
Not well	13 (8.7%)
Speaking Other Language	
Yes	106 (70.7%)
No	44 (29.3%)
Healthcare Coverage	
Yes	130 (87.2%)
No/not sure	19 (12.8%)
Insurance	
Private	86 (59.3%)
Public (Medicare, Medicaid, military)	40 (27.6%)
No insurance	19 (13.1%)
Religion	
No/not sure	29 (19.1%)
Yes	123 (80.9%)

*n* = 152.

**Table 2 ijerph-19-15917-t002:** Knowledge and Attitude Survey Question Response Rates.

Type	Question	No/Not Sure	Yes
Knowledge 1 *	Have you heard of the term ‘clinical trial’?	63 (41.5%)	89 (58.6%)
Knowledge 2	Do you agree or disagree with the following statement? “Clinical trials test how safe and useful a new drug is against cancer and other diseases”.	25 (16.5%)	127 (83.5%)
Knowledge 3 *	Does taking part in a clinical trial mean you might not receive the treatment being tested?	46 (30.3%)	106 (69.7%)
Knowledge 4 *	Do you agree or disagree with the following statement? In a clinical trial, the sponsor pays for the new drug being tested while all other costs are billed to your insurance company.	102 (67.1%)	50 (32.9%)
Attitude 5	Have you ever participated in a clinical trial?	144 (94.7%)	8 (5.3%)
Attitude 6	Has anyone you know taken part in a clinical trial?	128 (84.2%)	24 (15.8%)
Attitude 7	If you had cancer and were asked to be in a cancer clinical trial, would you take part?	40 (26.3%)	112 (73.7%)
Attitude 8	If you had cancer and were offered a cancer clinical trial, would you take part if it meant you needed to leave Guam for treatment?	36 (23.7%)	116 (76.3%)
Attitude 9	If you had cancer, would you prefer to take part in a cancer clinical trial offered in Guam rather than going off island for the same treatment?	32 (21.1%)	120 (78.9%)
Attitude 10	If you had cancer, do you believe you would benefit by taking part in a clinical trial in Guam?	31 (20.4%)	121 (79.6%)
Attitude 11	If you had cancer, would you take part in a clinical trial if you believed it would help other people in your community?	11 (7.2%)	141 (92.8%)
Attitude 12	If you had cancer, would you take part in a clinical trial if you believed it would lead to new treatments for cancer?	13 (8.6%)	139 (91.4%)
Attitude 13 *	Do you think you would receive good quality treatment from a clinical trial offered in Guam?	61 (40.1%)	91 (59.9%)
Attitude 14 *	Do you think that people who take part in cancer clinical trials are treated like ‘guinea pigs’?	99 (65.1%)	53 (34.9%)
Attitude 15	Would the possibility of serious side effects stop you from taking part in a clinical trial?	52 (34.2%)	100 (65.8%)
Attitude 16	Do you think your health insurance would cover you if you took part in a clinical trial in Guam?	117 (77.0%)	35 (23.0%)
Attitude 17 *	Do you think you would have to pay more out-of-pocket expenses if you took part in a clinical trial in Guam?	67 (44.1%)	85 (55.9%)
Attitude 18	If you had cancer, would you want your doctor to offer you a clinical trial in Guam?	32 (21.1%)	120 (78.9%)
Attitude 19	If you had cancer, would you be willing to change doctors in order to take part in a clinical trial in Guam?	32 (21.1%)	120 (78.9%)
Attitude 20 *	If you had cancer, would the doctor’s ethnicity be important in your decision to take part in a cancer clinical trial?	105 (69.1%)	47 (30.9%)
Attitude 21	Would you be willing to take part in a cancer clinical trial if the doctor was not your ethnicity?	38 (25%)	114 (75%)
Attitude 22 *	If your doctor gave you advice that goes against your cultural beliefs, would you listen to them?	56 (36.8%)	96 (63.2%)
Attitude 23	Do you trust western medicine?	30 (19.7%)	122 (80.3%)
Attitude 24	How important is having the emotional support of your family or social group in your decision to take part in a cancer clinical trial?	8 (5.3%)	144 (94.7%)
Attitude 25 *	How important is the support of your religious community if you decided to take part in a cancer clinical trial?	68 (44.7%)	84 (55.35%)
Attitude 26 *	If you had cancer, would you seek traditional healing practices?	50 (32.9%)	102 (67.1%)
Attitude 27 *	Have you ever gone to a *suruhano* for health care?	90 (59.2%)	62 (40.8%)
Attitude 28	Can a suruhano treat cancer?	129 (84.9%)	12 (15.1%)
Attitude 29 *	If you had cancer, would you seek treatment by a *suruhano*?	94 (61.8%)	58 (38.2%)
Attitude 30	Do you believe cancer is caused by *taotaomo’na* (ancient spirits)?	144 (94.7%)	8 (5.3%)
Attitude 31	Do you trust traditional medicine?	45 (29.6%)	107 (70.4%)

* Distribution of Survey Question for which combined categories of ‘Yes’ or ‘No/Not Sure’ comprised more than 30% of the sample: *n*(%). Data analyses were reported only for questions that met this analytical criterion.

**Table 3 ijerph-19-15917-t003:** Adjusted Odds Ratio and 95% Confidence Interval of Multivariate Logistics Regression.

Variable	Knowledge 1: Have You Heard of the Term ‘Clinical Trial’?	Knowledge 3. Does Taking Part in a Clinical Trial Mean You Might Not Receive the Treatment Being Tested?	Knowledge 4. In a Clinical Trial, the Sponsor Pays for the New Drug Being Tested while All Other Costs Are Billed to Your Insurance Company.	Attitude 13: Do You Think You Would Receive Good Quality Treatment from a Clinical Trial Offered in Guam?	Attitude 14: Do You Think That People Who Take Part in Cancer Clinical Trials Are Treated Like ‘Guinea Pigs’?	Attitude 17: Do You Think You Have to Pay More Out-of-Pocket Expenses If You Took Part in a Clinical Trial in Guam?	Attitude 20. If You Had Cancer, Would the Doctor’s Ethnicity Be Important in Your Decision to Take Part in a Cancer Clinical Trial?	Attitude 22: If Your Doctor Gave You Advice That Goes against Your Cultural Beliefs, Would You Listen to Them?	Attitude 25: How Important Is the Support of Your Religious Community if You Decided to Take Part in a Cancer Clinical Trial?	Attitude 26: If You Had Cancer, Would You Seek Traditional Healing Practices?	Attitude 27: Have You Ever Gone to a *suruhano* for Health Care?	Attitude 29: If You Had Cancer, Would You Seek Treatment by a *suruhano*?
Age											1.03 (1.01–1.06) *	0.97 (0.95–0.99) *
Gender (Ref: Male):												
*Female*										2.54 (1.16–5.54) *		
Ethnicity (Ref: Caucasian)												
*CHamoru*						5.34 (1.68–17.00) **			27.70 (3.47–221.00) **			
*Filipino*									42.20 (4.98–357.0) **			
*Other*												
Marital Status (Ref: Married/Living as Married)												
*Single*												
*Divorced/Widowed*										0.19 (0.06–0.60) **		
Education (Ref: ≤High School)												
*Some college or technical school*								3.42 (1.17–10.00) *				
*≥College graduate*	4.43 (1.97–9.95) ***		0.23 (0.07–0.74) *				0.34 (0.15–0.74) **	3.20 (1.52–6.75) **				
Personal Income (Ref: <$50,000)												
*≤$50,000*			6.53 (1.81–23.57) **	2.44 (1.18–5.02) *								
*Refused*			9.00 (1.84–44.11) **									
Born Country (Ref: USA)												
*Guam*											9.84 (2.95–32.75) ***	
*Philippines*					6.14 (1.84–20.50) **							
*Other*												
Employment Status (Ref: Employed)												
*Retired*		0.37 (0.15–0.93) *										
*Unemployed/student/homemaker/unable to work*		0.35 (0.15–0.85) *										
English Fluency (Ref: Not well):												
Well	6.02 (1.21–29.88) *		0.04 (0.01–0.25) ***									
Speaks Other Language (Ref: No):												
Yes	0.31 (0.13–0.76) *						3.40 (1.29–8.95) *					
Insurance (Ref: Private)												
*Public (Medicare, Medicaid, military)*			7.51 (2.65–21.3) ***									
*No Insurance*			6.90 (1.85–25.71) **									
Have Religion (Ref: No/not sure): Yes												

Adjusted odds ratio 
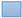
 < 1 
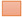
 > 1 * *p* < 0.05; ** *p* < 0.01; *** *p* < 0.001. Row percentage was presented.

## Data Availability

The data presented in this study are available on request from the corresponding author. The data are not publicly available due to protection of study participants’ privacy.

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
