# Peer review of "Knowledge and Attitudes of Guam Residents towards Cancer Clinical Trial Participation"

_ijerph, 2022, doi:10.3390/ijerph192315917_

Round 1

Reviewer 1 Report

This is an interesting study that addresses a gap in knowledge - namely attitudes of minority groups in taking part in clinical trials. The paper is well written and the discussion and conclusions are clear. My main comments relate to the methods and results which could be presented more clearly. In particularly, clearly separating out methods, results and discussion points. The file has comments relating to specific parts of the paper. 

Reviewer 2 Report

The study attempted to investigate the knowledge and attitudes of residents towards participation in clinical trials. The main challenge is the novelty of the study. It would be good if they could clearly show what this study adds to the body of knowledge. However, some comments follow that might improve the quality of the study.

 Abstract

Background: the justification for the necessity of this study wasn't convincing. The fact that there were no clinical cancer studies in the region in question, despite high mortality rates, should have prompted the authors to conduct one. Why then do they investigate perceived knowledge and attitudes? In my opinion, the background should be rewritten to convince the reader why this study was conducted.

Methods: I felt that more was needed in this section, and as a reader I should get an idea of what was done by just reading the summary. Please, provide the following information in the abstract:   1.       Instrument used for data collection (knowledge and attitude), and their psychometrics. 2.       The fact that you used logistic regression, please indicate if any adjustment were made e.g., adjusted odds ratio (AOR) at 95% confidence interval. 3.       Could the authors indicate what descriptive statistics were used in the analysis of the survey questions (e.g., frequency and percentage)? 4.       Please justify the why of the telephone survey.   Results

 1.       Could the authors present the result in such a way that it is short and concise? 

Example: One hundred fifty-two survey respondents included CHamoru (47.0%), Filipino (26.5%), Caucasian 15 (11.3%) and Other (15.2%). Could be best represented as: 152 people participated in the survey, most of them are CHamoru (n = xx, 47.0%).

2.       Other results might also follow the above pattern of presentation.

Introduction

1.       Could the authors exercise caution with this statement: “Therefore, research is needed to understand knowledge and attitudes that impact Pacific Islanders’ decision to participate in cancer clinical trials and increase their enrollment.”   2.       From the literature you have presented, lack of knowledge and awareness, cost, insurance coverage, mistrust, and fear of experimentation among others are some of the barriers to ethnic minority participation in clinical trials. You went further to justify that “To date, cancer clinical trials have not been offered in Guam and information on the knowledge and attitudes of its population towards such trials is limited.” You have already classified them as ethnic minorities to which the above factors might apply; what specifically do you want to explore that has not yet been covered? I did not find your argument convincing, to be precise. Should it be that you are interested in studying the relationship between socio-demographic factors and resident knowledge and attitudes? Please clarify.

Material and Methods    1.       If I understood the authors correctly, the survey was conducted using self-developed and unvalidated measures. Furthermore, the adopted items were those factors listed as barriers to ethnic minority participation in clinical trials that might potentially be study outcomes. Therefore, this should be considered the major limitation of the study, and readers should treat the results with caution. A pilot test alone could not make them a valid and reliable instrument.   2.       I couldn’t seem to understand why a Table 1 with results was placed under Methods.

3.       Although inclusion criteria were provided, I couldn't find information on consent and recruitment methods. Do they call any random numbers for participation?

4.       While it is fine for a third party to conduct the survey, it is not clear to readers what the proprietary random digit dialing software was used for.

Data Analysis   1.       Software – Did you use any? Please state its name, publisher and their location. 2.       Please clearly state the merged variables and why? Certain demographic categories were combined based on the distribution of responses for each variable to facilitate the data analysis. 3.       Data analysis must not be written like stories, please. If you have coded a variable, please provide the codes.   General comment In the interest of the readers, the authors should state the specific objectives of this study and present the results accordingly. The results presented seemed a bit chaotic. There were a number of questions in the discussion; however, answering the questions raised in Background and Methods might lead the authors to review them.    
